# Spermatocytic Tumor: A Review

**DOI:** 10.3390/ijms24119529

**Published:** 2023-05-31

**Authors:** Simona Secondino, Alessandra Viglio, Giuseppe Neri, Giulia Galli, Carlotta Faverio, Federica Mascaro, Richard Naspro, Giovanni Rosti, Paolo Pedrazzoli

**Affiliations:** 1Oncology Department, Fondazione IRCCS Policlinico San Matteo, 27100 Pavia, Italy; s.secondino@smatteo.pv.it (S.S.); gi.galli@smatteo.pv.it (G.G.); carlotta.faverio01@universitadipavia.it (C.F.); federica.mascaro01@universitadipavia.it (F.M.); p.pedrazzoli@smatteo.pv.it (P.P.); 2Anatomic Pathology Unit, Fondazione IRCCS Policlinico San Matteo, 27100 Pavia, Italy; a.viglio@smatteo.pv.it (A.V.);; 3Department of Molecular Medicine, University of Pavia, 27100 Pavia, Italy; 4Department of Internal Medicine and Medical Therapy, University of Pavia, 27100 Pavia, Italy; 5Urology Unit, Fondazione IRCCS Policlinico San Matteo, 27100 Pavia, Italy; r.naspro@smatteo.pv.it

**Keywords:** spermatocytic tumor, rare tumors, germ cell tumors

## Abstract

Spermatocytic tumor (ST) is a very rare disease, accounting for approximately 1% of testicular cancers. Previously classified as spermatocytic seminoma, it is currently classified within the non-germ neoplasia in-situ-derived tumors and has different clinical-pathologic features when compared with other forms of germ cell tumors (GCTs). A web-based search of MEDLINE/PubMed library data was performed in order to identify pertinent articles. In the vast majority of cases, STs are diagnosed at stage I and carry a very good prognosis. The treatment of choice is orchiectomy alone. Nevertheless, there are two rare variants of STs having very aggressive behavior, namely anaplastic ST and ST with sarcomatous transformation, that are resistant to systemic treatments and their prognosis is very poor. We have summarized all the epidemiological, pathological and clinical features available in the literature regarding STs that have to be considered as a specific entity compared to other germ GCTs, including seminoma. With the aim of improving the knowledge of this rare disease, an international registry is required.

## 1. Introduction

Testicular tumors are classified as germ-cell-origin and sex-cord-stroma tumors [1]. Germ cell tumors (GCTs) include seminoma and non-seminoma histologies; the latter encompasses teratoma (postpubertal type), embryonal carcinoma (EC), choriocarcinoma, yolk sac tumor (YST) and a mixture of these components, including seminoma. Pure histology is a rare event and frequently, non-seminoma are composed of multiple histologies, including seminoma components (so-called mixed germ cell tumors).

In 2020, GCTs represented the most frequent neoplastic disease among men aged 15 to 44 in 62 countries worldwide [2,3], with a slight predominance of seminoma in the fourth decade of life.

Spermatocytic tumor (ST) is a rare GCT derived from postpubertal-type germ cells and was previously called spermatocytic seminoma due to a false belief in its origin from germ cell neoplasia in situ [4].

Until 2016, spermatocytic tumor was called spermatocytic seminoma, but it included a distinct testicular tumor, having particular clinical-pathologic features when compared with other forms of GCTs, so the World Health Organization [4] changed the name of spermatocytic seminoma to spermatocytic tumor. Given the rarity of this neoplasm, there is a paucity of data in the literature evaluating this condition.

Differently from other GCTs, ST is not known to occur in extratesticular sites, and this tumor has an indolent course [5,6,7]. Most STs of the testis are considered benign tumors. As STs are extremely rare, there are limited data on the optimal management of patients having a localized or a metastatic disease. Nevertheless, there are two rare variants reported, having a very aggressive outcome: anaplastic ST and ST with sarcomatous transformation. They are highly resistant to treatments and the prognosis is very poor [8].

A web-based search of MEDLINE (PubMed) was performed in order to identify pertinent articles by using spermatocytic seminoma or spermatocytic tumor keywords.

This review was performed aiming to summarize the epidemiology and pathogenetic and histologic features, to provide the best available clinical data about recurrence rates of STs during follow-up and to discuss the available treatments and outcomes of patients with STs.

## 2. Epidemiology

ST is a rare testis tumor, accounting for about 1% of all testicular tumors [9]. Although its incidence has been reported at around 52–56 years of age, it has never been considered as a tumor of the elderly population [6,9,10,11,12]. Nearly 70% of patients are over 40 years of age, and it is virtually absent in children, adolescents and young adults, although the youngest patient described in the literature was 19 years old [9]. 

In an Australian series among 9658 cases of primary testicular tumors in the period 1982–2002, only 58 cases of STs were reported, and the Australian age-standardized incidence rate for spermatocytic seminoma was 0.4 cases per million. The mean number of cases diagnosed per year was 2.8, ranging from 0 to 7. In particular, the standardized incidence was 0.3 cases per million for men younger than 55 and 0.8 per million for men 55 years or older [9]. The mean patient age at diagnosis was 53.5 years, 50% of men were diagnosed at 54 years or younger, and approximately 25% of patients were younger than 40 years [9]. This cohort of patients reported not only the youngest patients with ST (19 years old), but also the oldest patient reported in literature, who was 92 years old [9]. In addition, the Australian series detected no statistical increase in the incidence of STs in the previous 20 years, but the ability of the authors to identify trends was limited by the small number of cases available [9]. 

In a more recent analysis reported from a large U.S. cancer registry from 2006 to 2016, of 53,481 adults who received orchiectomy, 29,208 were diagnosed with seminoma and 299 with STs, resulting in an incidence of 1% of all seminoma [12]. In this cohort, ST patients were more likely to be older and have medical comorbidities, larger tumors and less advanced disease on presentation. The median age at diagnosis in patients with ST was 56.6 years, compared with 38.5 years in classic seminoma, a finding consistent with previous Australian series. Some authors have also reported that 30% were from 30 to 39 years, 35% were from 40 to 49 years and 35% were 60 years or older [11].

In men aged 50 years and older, in a recent survey from 41 US cancer registries for the years 1999–2014, 16,735 patients with testicular tumors have been reported. In this series, 9353 were seminoma, 2227 were non-seminoma, 533 were spermatocytic tumors, 4334 were mainly non-Hodgkin lymphomas and 288 were sex cord stromal tumors [13]. The higher incidence (3%) of STs reported in this cohort reflects the peculiar prevalence of this disease in men aged 50 years and older. 

In countries with lower incidences of testicular cancer, the percentage of STs among GCTs has been reported as 1%, with no particular differences with data from countries and areas with higher incidence of testicular cancers [14]. In New Zealand, there is a “puzzling” incidence of testicular neoplasia especially within the Maori community, with a rate of 28/100,000, much higher than the rates among European-origin New Zealanders, Pacific Islanders living in the country and the areas of Micronesia or Melanesia. At the present time, we have no data regarding ST in this ethnic group [15]. 

## 3. Pathophysiology

Spermatocytic tumor was considered closely related to seminoma and classified as subtype of this disease from the time of its first description in 1946 by Masson, who named this entity as “le seminome spermatocitaire” [16]. 

The reason for the WHO classification change [4] was due to several important differences. First, STs have a different origin compared with other GCTs: they derive from more mature germ cells (likely premeiotic germ cells at a transition stage between spermatogonia and spermatocytes), in contrast to postpubertal-type germ cell tumors such as seminoma, which are thought to originate from primordial germ cells/gonocytes. Increased recognition of this distinction led to a revision in the nomenclature of these tumors from spermatocytic seminoma to ST, particularly to avoid overlapping terminology with GCNIS-derived seminoma. Second, unlike seminoma, STs do not exhibit gains of the short arm of chromosome 12, which is a frequent finding in (GCNIS-derived) postpuberal type germ cell tumors [17,18]. Third, STs show a unique amplification of chromosome 9, corresponding to the DMRT1 gene, not reported in all GCNIS-related tumors, but also in prepuberal teratoma and prepuberal YST [19]. The DMRT1 gene, located at the end of the 9th chromosome, is found in a cluster with two other members of the gene family (DMRT2, DMRT3), having in common a zinc finger-like DNA-binding motif (DM domain). The DMRT1 protein is located in the spermatogonia and in the spermatocytes, specifically in the nucleus, and its role is critical during embryogenesis and male germ cell maturation.

Figure 1 reported the different origins of GCTs and STs.

## 4. Histopathology

ST is a germ cell tumor, derived from postpubertal-type germ cells; tumor cells have a close resemblance to spermatogonia or early primary spermatocytes. Macroscopically, STs have sharp borders, the cut surface is usually grey-white and glistening, mucoid material may be present and foci of necrosis are hardly detected. STs usually display a multinodular or diffuse pattern; very rarely, they can present with a pseudoglandular, cystic, trabecular or nested pattern. The tumors are arranged in diffuse sheets and usually lack fibrous septa and lymphoid infiltrates. 

Neoplastic elements are heterogeneous. There are three types of cells: (a) small cells, 6–8 microns, with scant cytoplasm; (b) intermediate cells, 15–20 microns, with round nuclei and chromatin ranging from granular to spireme-type; and (c) giant cells, 50–100 microns, with multiple nuclei. 

Unlike GCTs, there is no germ cell neoplasia in situ (GCNIS), and no significant inflammatory infiltrate, granulomas, fibrovascular septa or cytoplasmic glycogen can be observed; these features may prove useful in the differential diagnosis with classic seminoma.

In the past, STs with a clear prevalence of nucleolated intermediate cells were termed as “anaplastic”; this description is currently not encouraged, as “anaplastic STs” seem to share the same biological behaviour as classical STs. Some authors have reported anaplastic ST in some tumors having parameters such as increased mitotic count (>30/10 HPF), the presence of prominent nucleoli, vesicular nuclei, bizarre giant forms, areas of necrosis and abnormal mitotic figures [20], but they in their reported series, did not find any cases that fit the given criteria [7]. 

ST undergoing sarcomatous transformation, which occurs in approximately 6% of cases, is a rare but well-documented event [11]. The sarcomatous component can be either intermixed with classic areas of ST or may be a separate nodule lying adjacent to the classic ST. The sarcomatous component is usually an undifferentiated spindle cell sarcoma, but sarcoma with rhabdomyosarcoma and chondroid differentiation have also been reported [7]. As documented in the literature, ST with a sarcomatous component has the same age incidence as pure ST [21]. 

Amid the published series, 21 patients were reported with sarcomatous transformation [21], usually showing a sudden rapid growth in the size of the tumor. The histology was undifferentiated sarcoma in 6 patients, spindle cell sarcoma in 5 patients, rhabdomyosarcoma in 8, chondrosarcoma in 1 and chondrosarcoma plus rhabdomyosarcoma in 1 [21]. Due to the aggressive behaviour of the disease, with a significant risk for metastatic spread, the recognition of such a component is extremely important, warranting adjuvant treatment after surgery. Nevertheless, STs with sarcomatous transformation are highly resistant to cytotoxic chemotherapy, and patients affected with such STs reported a median survival of 6 months [21]. In these cases, the residual ST component, usually scarce, may be either admixed with, or at the edge of, the sarcoma elements. The main differential diagnoses are primary sarcoma of the paratesticular region and a secondary somatic malignancy arising in a teratoma. Adequate sampling in cases of large-sized tumors may help in recognizing areas of classic morphology of ST. The reason for the emergence of sarcoma in ST is still not fully understood, but some authors have suggested it can be explained by an anaplastic transformation or dedifferentiation of a well differentiated ST [7]. 

The differential diagnosis of ST should be made with seminoma, particularly with microcystic patterns [22], embryonal carcinoma [23] and malignant lymphoma [24,25]. In each case, both morphological and immune phenotypical features are to be considered.

Seminoma is characterized by homogeneous neoplastic cells; fibrovascular septa are usually present, as well as a chronic inflammatory infiltrate, possibly comprising granulomas; and GCNIS foci are easily spotted. As with ST, seminoma is CD117/ckit+; however, typical seminoma is reactive with GCT markers, such as OCT3/4, PLAP and D2-40. In contrast, increased immunohistochemical reactivity for FGFR3, HRAS and DMRT1 has been shown in ST [26].

Embryonal carcinoma has markedly pleomorphic tumor cells, and a variety of architectural patterns (glandular, cystic, papillary) are rarely observed in ST. Despite the “anaplastic” nature of this uncommon ST variant, they still retain the round-nuclei characteristic of this tumor, whereas those of embryonal carcinoma are more irregular in almost all instances. The typical staining for CD30/BerH2 and OCT3/4 allows the diagnosis of embryonal carcinoma [23]. From the clinical perspective, embryonal carcinoma occurs at an even younger age than does the usual case of typical seminoma, on average occurring a decade earlier, peaking between 25 and 35 years of age.

High-grade non-Hodgkin lymphomas, such as an anaplastic variant of diffuse large B-cell lymphoma and anaplastic large-cell lymphoma, also need to be included in the differential diagnosis, especially in elderly patients. A variety of immunohistochemical differences between lymphoma and ST exist and should be applied if this differential occurs. The positivity for an ICH panel for lymphoid markers, including CD45/LCA, CD20 and CD3 is not observed in STs [24,25]. 

STs are negative for many GCT markers, such as OCT3/4, D2-40/podoplanin, PLAP, alfa-fetoprotein, glypican-3, hCG, CD30/BerH2, whilst they are positive for CD117/ckit and SALL4, which are commonly used stains in pathology routines (Table 1), and for a range of antigens expressed in spermatogonia and early spermatocytes (MAGE-A4, OCT2, SSX, SAGE1, SYCP1, XPA, FGFR3, DMRT1, HRAS) [27].

## 5. Clinical Features

STs usually present with a unilateral mass. Bilateral involvement has been reported, even though is more often documented in ST compared with other germ cell tumors [11,25,28,29]. One of the large series published showed bilateral STs in 5% of testicular tumor; one patient (1%) had synchronous bilateral mass in undescended testes at age 67, and 2 patients (3%) had metachronous bilateral testicular masses at intervals of 10 and 16 years, respectively [11].

Variable tumor size has been recorded, ranging from 2 to 20 cm with an average of 7–8 cm [7,30]. Nevertheless, though ST patients are more likely to have larger tumors on presentation with respect to seminoma, they are less likely to have <pT2 stage, and only 16.4% of STs are smaller than 3 cm, compared with 27.6% in the seminoma group [12]. In this cohort, patients with ST were nearly twice as likely as classic seminoma patients to have tumors larger than 3 cm. While tumor size is generally greater in ST, these data do not translate into greater locally advanced or metastatic disease. Indeed, patients with ST are also less likely to have regional nodal involvement (clinical stage II) or distant metastases on presentation, compared with seminoma [12]. Specifically, it has been reported that nodal involvement is identified in only 3.7% of ST patients, and metastatic disease at presentation in only 0.7% cases [12]. 

Due to the indolent nature of the tumor, there is often some appreciable interval from the time of initial recognition of a lesion by the patients to their seeking medical attention, and so most patients developed large testicular mass before diagnosis [11]. In a few cases, the diagnosis was confirmed after the work-up of other symptoms including infertility, metastases, hydrocele, back pain or weight loss [31]. Some authors reported that in about one-third of the cases, a mass had been known for about 12 months or longer [32], sometimes even for 5 years or more. 

The association of ST with cryptorchidism is extremely rare, as only two cases are reported in the literature so far [11,33]. Cryptorchidism is a well-known risk factor for testicular cancer [34], but does not appear to be a risk for ST. No reported case had an association with either changes in testosterone or estrogen levels, or clinical signs of gynecomastia [31]. 

While serum tumor markers such as gonadotropin (beta-HCG), alpha-fetoprotein (alfaFP), and lactate dehydrogenase (LDH) are expressed in about 50–60% of GCT patients [35], they are reported as increased in less than 1% of ST patients [31]. Sonography details were infrequently reported, such as calcification in one patient and heterogeneous appearance in another [31].

STs never arise in extragonadal sites such as the anterior mediastinum, retroperitoneum, pineal gland and sacrococcygeal area, and there is no ovarian homologue, as is the case for seminoma (so-called dysgerminoma), although one case has been described in a dysgenetic gonad of a phenotypic female who had an associated gonadoblastoma [36]. 

Table 2 describes differences between seminoma and spermatocytic tumor. 

## 6. Treatment

Surgical orchiectomy is the standard treatment for STs, as testis-sparing techniques are limited by the difficulties for the pathologist in differentiating ST from seminoma GCT in a frozen section [37]. 

As STs rarely metastasize, surgery alone is the standard of care. In the past, adjuvant radiotherapy was performed, but it has gone into disrepute in recent studies [7,38]. 

In a recent systematic retrospective review of 146 cases [11], only six patients (4%) with STs received adjuvant therapy: radiotherapy or chemotherapy or both. Three patients received platinum-based chemotherapy and only one patient was disease-free at follow-up. Two patients underwent radiotherapy to the pelvis: one patient died 1 year later of gastric lymphoma, and the other patients after “20 sessions of radiotherapy” did not report any disease recurrence. Only one patient received both adjuvant radiotherapy and chemotherapy, and remained disease-free at 2 months [11].

A large retrospective analysis of testicular cancer patients was conducted in the United States from 2006 to 2016 [12]: among 53,481 patients receiving orchiectomy, 29,208 (54.6%) were diagnosed with classic seminoma and 299 (1%) had STs. Among ST patients, about 80% had a local disease and only 3.7% had regional nodal involvement on presentation, while 0.7% had metastatic disease at diagnosis. With regard to post-surgery management, whilst 47.6% of patients with seminoma underwent surveillance, 73.6% of patients with ST did not receive adjuvant treatment. Notably, the adjuvant therapy was directly associated with clinical stage. When stratified by year, there was an increase in patients undergoing surveillance after orchiectomy for both histologies. Concerning ST, patients undergoing surveillance increased from 41% in 2006 to 93% in 2016, while patients undergoing radiotherapy (within 90 days after orchiectomy) decreased from 47% in 2006 to 7% in 2016 [12]. These data reflect what happened in seminoma patients, in which radiotherapy also declined to <10% by 2016, due to the nearly twofold risk of secondary malignancies in seminoma patients who received radiotherapy [39].

Metastases during follow-up were diagnosed after a median follow-up of 5.5 months (range 2–21 months), and the metastatic relapse was reported at the retroperitoneal lymph nodes, lung, liver and brain [31]. It is worth noticing that this high incidence reported in the literature is likely positively affected by the publication of rare and atypical clinical findings. It is possible that the real incidence of metastatic disease in ST is less than 1%, as reported by Patel et al. [12]. In patients treated for metastatic spread, durable response after retroperitoneal lymph node dissection or surgical metastasectomy was observed in 4% of patients; complete response to chemotherapy was reported in 4% of ST patients [40]. 

Due to the rarity of metastatic spread, considering the data reported in the literature, surveillance remains an avenue for improving patient care for men with ST, as most of these patients likely do not require aggressive initial measures [6,8,12,31]. 

The US study also compared survival outcomes for men with ST relative to classic seminoma. After adjusting for differences in baseline demographics, such as age and comorbidity, as well as stage distribution, they observed no differences in overall survival between the two groups [12]. The favorable survival outcomes for ST, despite a reduction in the use of adjuvant therapy after orchiectomy over time, is encouraging. 

On the other hand, STs with sarcoma transformation or anaplastic subtypes showed metastatic disease in 45% and 29% of cases, respectively; compared to ST, patients having these histological variants were more likely to have metastatic disease. These subtypes are highly resistant to cytotoxic chemotherapy and they represent a very aggressive disease: patients have a poor prognosis, with a median survival of 5 months [31,41]. Despite the high risk of metastases, the guideline of treatment for STs with sarcomatous transformation has yet to be established, albeit adjuvant chemotherapy seems to be a possible choice. In the literature, five of these patients received platinum-based chemotherapy, and all responded poorly and died shortly (within 3–14 months) after diagnosis [12]. However, only one case of metastatic ST responded well to the VIP combination chemotherapy after radical orchiectomy [42]. 

## 7. Conclusions

STs are a very rare entity of germ cell neoplasia, with just a few hundred cases reported so far.

These tumors have distinctive clinical and pathological features which allow their accurate diagnosis and management. 

Their natural history tends to be rather indolent with an excellent prognosis, and, although STs are, by definition, not associated with GCNIS, due to the large average tumor size and absence of limited testis-sparing surgery, orchiectomy alone is the standard approach for the vast majority of cases. In any case, orchiectomy, rather than testis-sparing surgery, should also be standard of care because of the difficulty in differentiating ST from pure seminoma in frozen section analysis. 

The rare presence of anaplastic or sarcomatoid components is the major factor affecting metastatic spread, and, rarely, primary extragonadal disease may be present at diagnosis.

Based on the findings that all patients with recurring metastatic aggressive disease were diagnosed within the first two years (range 2–21 months), follow-up with CT scan could be probably be limited to the first 2 years. Assuming that only 7% of patients with STs may develop distant recurrences during follow-up [31], even if the real incidence seems to be much lower, attesting to 1% [12], the number of patients needing periodic imaging during follow-up is very limited. 

In the absence of large and/or prospective studies, no particular advice regarding adjuvant therapy can be given for STs; until more evidence becomes available, some authors suggest that radiotherapy or chemotherapy after orchiectomy should be considered in patients with poor prognosis histology, such as a sarcomatous element in ST or anaplastic subtypes. 

To improve the status quo in the future, a multicenter registry has been recently activated with the aim of collecting data from patients with rare testis cancer histologies, and thus to acquire more conclusive recommendations regarding clinical management and follow-up of these rare tumors, to be transferred to clinical practice [31]. 

## Figures and Tables

**Figure 1 ijms-24-09529-f001:**
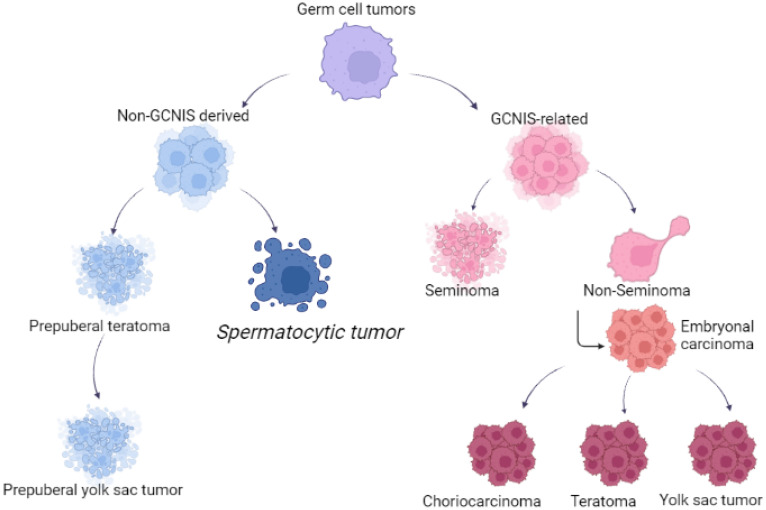
Different origin of germ cell tumors and spermatocytic tumors.

**Table 1 ijms-24-09529-t001:** Immunohistochemical differences between spermatocytic tumor and seminoma.

	Spermatocytic Tumor	Seminoma
CD117/ckit	+	+
D2-40/podoplanin	−	+
OCT3/4	−	+
PLAP	−	+
FGFR3	+	−
HRAS	+	−
DMRT1	+	−

**Table 2 ijms-24-09529-t002:** Clinical features and pathological aspects of spermatocytic tumor and seminoma.

	Spermatocytic Tumor	Seminoma
**Median Age (years)**	54	35
**Frequency of all GCT (%)**	<1%	60%
**Presence of GCNIS**	No	Yes
**Extragonadal site (% of all GCT)**	No	2–5%
**Expression of isochromosome 12p**	Absent	Present
**Sarcomatous component**	5–8%	No
**Bilaterality (synchronous and metachronous)**	8–10%	5%
**Stromal inflammatory reaction**	Absent	Present
**Association to cryptorchidism**	No	Yes
**Overall Survival at 5-y for Clinical Stage I (%)**	95%	99%
**Overall Survival at 5-y for advanced disease (%)**	Poor	72–86% *

GCT = germ cell tumors; GCNIS = germ cell neoplasia in situ. * according to IGCCCG (International Germ Cell Cancer Collaborative Group).

## Data Availability

No new data were created or analysed in this study: data sharing is not applicable to this article.

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
