# Peer review of "Spermatocytic Tumor: A Review"

_ijms, 2023, doi:10.3390/ijms24119529_

Round 1

Reviewer 1 Report

The topic is present in the literature, the narrative review is well organised, I'll change/delete the chapter ""advanced disease", the concept il well explained in many other parts of the text. 

I'll delete/make smaller fig.1 and add a table/figure to summarise the molecular and histopatological differences in detal like table 1 which is very clear. Some minor changes and advices can be found in notes in pdf attached, but the work doesn't need any major revisions. 

I'll also integrate some recente reviews that you dont' mention, like: 

- 10.1097/PAP.0000000000000302 

English is good but some sentences must be re-organised in order to be more clear. 

Author Response

I'll change/delete the chapter ""advanced disease", the concept il well explained in many other parts of the text. 

I have modified the text, as required  by the Reviewer. 

I'll delete/make smaller fig.1 and add a table/figure to summarise the molecular and histopatological differences in detal like table 1 which is very clear. 

I have modified the fig 1 and I have added a new table, as suggested (called table 1)

Some minor changes and advices can be found in notes in pdf attached, but the work doesn't need any major revisions. 

I have modified the text, according the suggestions of the Reviewer. 

I'll also integrate some recente reviews

I have added the review suggested. 

Reviewer 2 Report

Dear authors,

I congratulates for this original and well-articulated Review, which responds to the objectives set summary of the best current literature knowledge about the spermatocyte tumor.

The text is fine written, clear, and the readability is enjoyable for the reader. In addition, graphic illustration and table 1 conform to the text simplifying the concepts presented.

In my opinion this work can be improved before a final publication, following a minor revision:

It might be interesting for the reader to get an updated overview on the latest in vitro results, that show the effects of antidiabetic and natural compounds treatment on GCT models. Please see recent works about:

-         doi: 10.3389/fendo.2022.1051988.

-        doi: 10.3390/nu14112323.

With best regards,

Author Response

It might be interesting for the reader to get an updated overview on the latest in vitro results, that show the effects of antidiabetic and natural compounds treatment on GCT models. Please see recent works

Thank you for providing these two papers, showing the in vitro results on GCT models, but there are no data regarding spermatocytic tumors. 

Reviewer 3 Report

The review on "spermatocytic tumor" by Secondino et al is a very comprehendible and sound summary of the current data and knowledge on STs. The background of STs, as well as its specialties and pitfalls are nicely summarised. Also, focus is being put on the differentiation to other types of GCTs as well as ST variants with poor prognosis.  

There are only minor typos that may need to be corrected. I am just unsure if one point in table 1 is correct (ST = stromal inflammatory reaction present, SE = stromal inflammatory reaction absent). I have the impression that this may be mixed up. 

Author Response

I am just unsure if one point in table 1 is correct (ST = stromal inflammatory reaction present, SE = stromal inflammatory reaction absent). I have the impression that this may be mixed up

Dear Reviewer, thank you for your comments regarding the table, that has been promptly change according to your suggestion. 

I'm sorry for the uncorrect data.

Reviewer 4 Report

The authors have presented a review on spermatocyte tumors, focusing on the histopathological features that restrict their diagnosis. The article also highlights lesser-known clinical and cellular characteristics of ST subtypes as compared to seminomatous tumors and GCTs in general. Finally, the authors emphasize the limitations in treating advanced ST. However, there are several comments that need to be addressed before the publication of the manuscript.

I have a few suggestions:

1.     The article shows repetitive information in the introduction section about the particularities of the spermatocyte tumor, and its name changes by the WHO, which are mentioned during the text, with special attention in section 3. As well as the sites of metastases during section 7. I suggest condensing the information about the name changes which were prompted by misinformation about their origin and including it only in section 3. 

2.     The first two paragraphs of the introduction require citations. 

3.     The description in Figure 1 would benefit from improvement. To ensure consistency with the text describing histology subtypes based on cell origin, I urge the authors to include "derived/related" next to Non-GCNIS and GCNIS. 

4.     It is suggested to the authors to encourage the revision of the redaction in sections 3 and 5, in order to reduce the usage of words and implement more concise sentences, for example, in the first paragraph of section 5. 

5.     In section 4, the first three paragraphs pertaining to Histopathology could be merged to get one. 

6.     In paragraph 10 the Authors referred to "Fig 2" but there's only Fig 1. Given the potential to streamline the article with an integrative figure, I suggest improving Figure 1 to be applicable to cite it in all sections. 

7.     In the same paragraph, citations are required for "increased IHQ reactivity for FGFR3, etc". Additionally, citations are fundamental for paragraphs 11, 12, and 13.

8.     I suggest to the authors to consider revising the organization of the article to improve the flow of information and make it easier for the reader to follow along: 5. Clinical features could be moved instead "3. Pathophysiology" 

9.     To describe the i12p mutation, the authors could use the words Presence or Amplification instead of "Expression" in table one. 

Minor corrections on section 6: 

1.     Please Correct the typos “Test s” and “no adjuvant treatment”

2.     The use of the verb “mirror” does not seem adequate.

3.     Notice the constant use of certain words in reduced text space, such as “paucity”, and “among”. 

4.     Section 7 could be integrated into the text, instead of being a separate topic. 

5.     The conclusion could be summarized, particularly in paragraphs 5 and 6, in which its ideas were developed during the text.

Although the article is generally well written and easily understood, it is suggested that the authors review the comments and suggestions that I put above, in order to make the message and information clearer for the readers.

Author Response

Point 1: The article shows repetitive information in the introduction section about the particularities of the spermatocyte tumor, and its name changes by the WHO, which are mentioned during the text, with special attention in section 3. As well as the sites of metastases during section 7. I suggest condensing the information about the name changes which were prompted by misinformation about their origin and including it only in section 3.

We have modified the text according to your suggestions. 

Point 2: The first two paragraphs of the introduction require citations

We added new citations.

Point 3:  The description in Figure 1 would benefit from improvement. To ensure consistency with the text describing histology subtypes based on cell origin, I urge the authors to include "derived/related" next to Non-GCNIS and GCNIS.

We have corrected the figure 1 

Point 4:  It is suggested to the authors to encourage the revision of the redaction in sections 3 and 5, in order to reduce the usage of words and implement more concise sentences, for example, in the first paragraph of section 5

we have done some revisions of the redaction in section 3 and 5. 

Point 5: In section 4, the first three paragraphs pertaining to Histopathology could be merged to get one.

The three paraghaphs have been merged to get one. 

Point 6: In paragraph 10 the Authors referred to "Fig 2" but there's only Fig 1. Given the potential to streamline the article with an integrative figure, I suggest improving Figure 1 to be applicable to cite it in all sections.

We have corrected our mistake

Point 7: In the same paragraph, citations are required for "increased IHQ reactivity for FGFR3, etc". Additionally, citations are fundamental for paragraphs 11, 12, and 13

We added citations

Point 8:  I suggest to the authors to consider revising the organization of the article to improve the flow of information and make it easier for the reader to follow along: 5. Clinical features could be moved instead "3. Pathophysiology

Following your suggestion, which is also the one pointed out by another Reviewer, we have revised the text. 

Point 9: To describe the i12p mutation, the authors could use the words Presence or Amplification instead of "Expression" in table one

We have changed the word "expression", as required. 

Minor corrections on section 6: 

Point 1: Please Correct the typos “Test s” and “no adjuvant treatment”

We have corrected the text. 

Point 2: The use of the verb “mirror” does not seem adequate.

We have modified the term "mirror" with "reflect"

Point 3: Notice the constant use of certain words in reduced text space, such as “paucity”, and “among”. 

We check the text and reduce the words "paucity" and "among"

Point 4: Section 7 could be integrated into the text, instead of being a separate topic. 

We have modified the text, integrating section 7 into previous sections

Point 5: The conclusion could be summarized, particularly in paragraphs 5 and 6, in which its ideas were developed during the text.

We have modified the conclusions, according to Reviewer's suggestions. 

Round 2

Reviewer 4 Report

The authors reviewed and paid attention to the suggestions. I would only suggest that they improve the resolution of the figure. I have no more comments. No further comments.

Author Response

The authors reviewed and paid attention to the suggestions. I would only suggest that they improve the resolution of the figure. I have no more comments. No further comments.

I've modified the figure 1, according to the Reviewer suggestion
